# SEM Observation of the Filter after Administration of Blinatumomab: A Possibility of Leakage during Home Administration Using a Portable Infusion Pump

**DOI:** 10.3390/ijms24065729

**Published:** 2023-03-17

**Authors:** Megumi Takano, Motoki Inoue, Yuko Ikeda, Hidenori Kage, Tohru Inokawa, Kazuhiko Nakadate, Takeo Yasu, Yasumasa Tsuda, Kazumi Goto

**Affiliations:** 1Department of Pharmacy, St. Luke’s International Hospital, 9-1, Akashi-cho, Chuo-ku, Tokyo 104-8560, Japan; 2Department of Molecular Pharmaceutics, Meiji Pharmaceutical University, 2-522-1, Noshio, Kiyose, Tokyo 204-8588, Japan; 3Department of Basic Science, Educational and Research Center for Pharmacy, Meiji Pharmaceutical University, 2-522-1, Noshio, Kiyose, Tokyo 204-8588, Japan; 4Department of Medicinal Therapy Research, Pharmaceutical Education and Research Center, Meiji Pharmaceutical University, 2-522-1, Noshio, Kiyose, Tokyo 204-8588, Japan

**Keywords:** blinatumomab, leakage, continuous infusion, bispecific T-cell engagers, polysorbate 80, filter

## Abstract

Blinatumomab (Blincyto^®^ injection solution) is classified as a bispecific T-cell engaging (BiTE) antibody and is intended for the treatment of relapsed/refractory acute lymphoblastic leukemia. It requires continuous infusion to maintain therapeutic levels. Therefore, it is often administered at home. Monoclonal antibodies, which are administered intravenously, have the potential to leak depending on the nature of the administration devices. Therefore, we investigated device-associated causes of blinatumomab leakage. We observed no apparent changes to the filter and its materials after exposure to the injection solution and surfactant. From scanning electron microscopic images, precipitate on the surface of the filters was observed after physical stimulation of the injection solution. Therefore, physical stimulations should be avoided during the prolonged administration of blinatumomab. In conclusion, the findings of this study assist in the safe administration of antibodies using portable infusion pumps, taking into consideration the composition of drug excipients and the choice of filter type and structure.

## 1. Introduction

Blinatumomab (Blincyto^®^), classified as a bispecific T-cell-engaging (BiTE) antibody, selectively binds to the CD3 and CD19 sites on the target T and B cells, respectively. This drug is used to treat acute lymphoblastic leukemia in pediatric and adult patients [1,2,3]. Blinatumomab has a short half-life of approximately 2 h in the bloodstream and requires a continuous infusion to maintain therapeutic levels. The package insert for Blincyto^®^ in the U.S. recommends that it be initiated in the hospital to monitor for side effects such as infusion reactions, cytokine release syndrome (CRS), neurotoxicity, and tumor lysis syndrome (TLS) [4]. A lower probability of side effects would facilitate the intravenous administration of this drug using a portable precision infusion pump at the home of the patient. Consequently, the patients can continue with their daily activities during drug administration [5]. However, intravenous drug administration of blinatumomab at home is limited owing to issues such as preparation, administration, monitoring, and coordination of care. Notably, blinatumomab administration may be hindered owing to the low battery of the pump, the presence of air bubbles in the administration apparatus, and catheter stenosis [5,6,7,8]. Therefore, adequate education should be provided to the staff, patients, and their caregivers to facilitate the smooth functioning and in-house administration of the drugs.

In Japan, the CADD Legacy^®^ PCA pump (ICU Medical, Inc. San Clemente, CA) connected with the CADD^®^ extension set with a 0.2-micron air-eliminating filter (ICU Medical, Inc. San Clemente, CA) is often used for intravenous drug therapy at home (Figure 1a). Although the infusion filters are compatible with blinatumomab, there have been reports of liquid drug leakage from the air vent in the filter [9]. St. Luke’s International Hospital utilizes the JMS^®^ Infusion Filter Set (JMS Co., Ltd., Hiroshima, Japan) with a hollow-fiber inline filter (Figure 1b). Blinatumomab has been administered on five separate occasions without any incidences of leakage from the filter. We speculated that investigating the cause of the drug leakage or the lack thereof will facilitate the application of intravenous drug therapy at home. The package insert of the infusion filter set warns that the drug may leak if the surfactant is an excipient [10,11]. Surfactants cause the hydrophobic membrane of the filter to become hydrophilic, thereby increasing their permeability [12,13].

In this study, we aimed to choose an optimal structure for the filter, which would facilitate the uninterrupted administration of blinatumomab. Polysorbate 80 (PS80) plays an important role as a solubilizer and stabilizer that prevents absorption on the surface of a material and protects materials from interfacial stress [14]. However, PS80 is also known as a hydrophilizer of hydrophobic membranes [13]. As Blincyto^®^ contains PS80 as a stabilizer, we hypothesized that PS80 was a cause of infusion leakage. Therefore, the materials of the filter were immersed in PS80 solution, and the membrane strength after immersion was evaluated. The surface morphology of the filter was comprehensively examined after the liquid was poured using scanning electron microscopy (SEM). This study clarified the importance of accounting for the composition of the excipients in the medicine, as well as the selection of the filter type and structure.

## 2. Results

### 2.1. Effect of Surfactant on Membrane Strength

To ascertain the membrane strength by the adsorption of surfactants, the tensile strength of the membrane after immersion into PS80 was measured. Figure 2a shows the stress–strain curve at a constant speed. The tensile strength increased curvilinearly with an increase in tensile distance, breaking at the elongation point at 4.6 mm and a strength of 1.63 kg. Figure 2b displays tensile strengths at the various breaking points against time immersed in different concentrations of PS80. These results indicate that membrane strength is not influenced by the concentration of PS80 or immersion time.

### 2.2. Effect of the PS80 Solution on Liquid Leakage from the Filter during Mock Administration

To clarify the interaction between the PS80 solution and the filter, the mock solution was administered using similar concentrations as that used for PS80 and subsequently with 10× higher concentration. We did not observe any leakage from the filter under all conditions, nor did we observe any change in the color of the cobalt chloride paper. Furthermore, SEM observation of the filter membrane also revealed no significant change after PS80 administration (Figure 3).

### 2.3. Effect of Blincyto^®^ Administration

We first observed the hollow membrane, which was removed from the filter using SEM. Figure 4 shows the deposits observed on the hollow-fiber filter surface during blinatumomab administration. After administering blinatumomab for 96 h, we did not observe any leakage of the liquid; however, several precipitates were observed on the membrane surface (Figure 4).

### 2.4. Effect of Physically Stimulating the Blinatumomab Solution

To investigate the cause of blinatumomab aggregation, the drug was subjected to various conditions, such as changing temperature, ultra-violet (UV) light, and vigorous vibration. Figure 5 shows the solution after filtration under these conditions. We did not observe any precipitate at 40 °C after 15 min of filtration. However, an increased amount of precipitates were observed at 100 °C. Black light irradiation also generated precipitates; however, precipitates were not observed after UV irradiation and vibration.

## 3. Discussion

The results of this study indicate that the PS80 contained in Blincyto^®^ was not related to its leakage and the decreased mechanical strength of the filter membrane. The cause of leakage during long-term administration using a portable infusion pump may be attributed to filter blockage and increased internal pressure owing to protein aggregation. Furthermore, we confirmed that the PS80 concentration and immersion time did not affect the membrane strength, nor did they alter the filter surface. Therefore, PS80 was not directly related to the liquid leakage. SEM observations of the filter removed from the JMS^®^ Infusion Filter Set after blinatumomab administration revealed precipitates on the filter surface. Elementary analysis indicated that the precipitates contained organic compounds, suggesting that blinatumomab may have precipitated on the filter surface during continuous infusion.

Blinatumomab is a BiTE antibody, which possesses similar characteristics to monoclonal antibodies, which form aggregates under various conditions, including mechanical, chemical, and thermal stress, which can affect their biological activity [15,16,17]. It is difficult to avoid these stimuli during continuous infusion as conditions change in daily life. Therefore, aggregates were likely formed and deposited on the filter surface owing to the stress caused by various daily activities and the living environment. Therefore, we examined several stimulating conditions during which blinatumomab precipitates and observed that temperature was a relevant factor. Previous reports indicated that temperature is involved in increasing the aggregation rate, with monoclonal antibodies partially or completely unfolding at high temperatures and becoming destabilized to form non-covalent aggregates [16,17,18,19,20]. Concordantly, the JMS^®^ Infusion Filter Set filter-based administration used in our hospital resulted in no visible leaks. This could be attributed to the different structure and function of the filter compared with that in the CADD^®^ extension set with a 0.2-micron air-eliminating filter [12,13]. Inline filters can be divided into two types: hollow fiber and flat membrane. The hollow-fiber type has a larger membrane area as it is fitted with many thin tubular membranes. Furthermore, the CADD^®^ extension set consisted of only an air-vented filter, whereas the JMS^®^ Infusion Filter Set contained a hydrophilic membrane filter with an air-vent, the latter possessing a larger surface area. These differences suggest that deposition on the filter surface caused clogging of the membrane in the CADD^®^ extension set, thereby increasing the internal pressure and resulting in leakage. Blinatumomab may likely form aggregates owing to thermal stresses, which differ from the recommended storage conditions during home administration. This study has some limitations. Firstly, this study focused on blinatumomab as a BiTE antibody. Other antibody formulations are likely to contain additives other than PS80, which can adversely affect the filter material. Additionally, the types of filters and materials are not fully covered. Finally, insufficient consideration has been given to the physical stimuli that contribute to precipitation. However, we discovered a filter without shortcomings that is convenient for patients and caregivers in home administration, thus the urgent need to report it in the literature. Further comprehensive investigation of the temperatures under which precipitation may occur is required to estimate the likelihood of precipitation during home administration.

## 4. Materials and Methods

### 4.1. Materials

PS80 (Fujifilm Wako pure chemical, Osaka, Japan), distilled water (Otsuka Pharmaceutical Co., Ltd., Tokyo, Japan), and saline solution (Otsuka Pharmaceutical Co., Ltd., Tokyo, Japan) were used as received. Polyether sulfone (PES membrane, pore size 0.2 μm, Toyo Roshi Kaisha, Ltd., Tokyo, Japan) was used as the membrane. A JMS^®^ Infusion Filter Set (JMS Co., Ltd., Hiroshima, Japan) and CADD^®^ extension set with a 0.2-micron air-eliminating filter (ICU Medical, Inc, San Clemente, CA, USA) were used as infusion roots.

### 4.2. Effect of Surfactant Immersion on Membrane Strength

PS80 solutions were prepared at varying concentrations [0.1% (*w/v*), 1% (*w/v*), and 5% (*w/v*)]. PES membranes (105 × 2 mm) were immersed in these solutions for 24, 48, and 72 h. After being removed from the solution, the strength of the membrane was measured using a TA-XT plus Texture analyzer (Stable microsystems, Godalming, UK). The specimens were clasped using a clip-shaped jig, fixing the vertical interval at 80 mm. The upper jig was pulled up at 2 mm/s, and the relationship between the force and distance was recorded until the breaking point was reached.

### 4.3. Effect of Surfactant Concentration and Exposure Time on the Filter

Approximately 250 mL of 0.003% (*w/v*) and 0.03% (*w/v*) PS80 saline solution was inserted into a CADD^®^ pump infusion set. As shown in Figure 1, the lines were connected to the same condition as actual clinical conditions, and priming was performed with 20 mL of the same solution. CADD^®^ extension set and JMS^®^ Infusion Filter Set were used (Figure 1a,b). The settings for the CADD Legacy^®^ PCA pump were as follows: a dose rate of 2.5 mL/h, a delivered volume of 0 mL, air bubble detection set to “off”, upstream occlusion detection set to “off”, and downstream occlusion detection sensitivity set to “standard”. The cassette and CADD Legacy^®^ were hung with the filter placed at the same height as the tip of the line. An 18G needle was connected to the tip of the line, and a 96 h mock administration was performed into an empty infusion bag. Liquid leakage from the administration route was visually confirmed, in addition to observing changes in the color of the cobalt chloride paper. After simulating the administration, the filter portion was fixed with formalin and dried at a critical point with ethanol and tert-butyl alcohol. The filter was coated in platinum and then observed using SEM (JCM-7000, JEOL, Ltd., Tokyo, Japan) [21].

### 4.4. Effects of Physical Stimulation on Blinatumomab

After dissolving blinatumomab in 3 mL of distilled water, the following stimulation conditions were applied to the drug solution: an increase in the temperature (40 °C 15 min, 100 °C 15 min), vibration for 1 min (VORTEX GENIE2, M&S Instruments Inc., Osaka, Japan), UV light irradiation (30 W 15 min), and black light exposure (3 W 15 min). Before SEM observation, the solutions were passed through a filter. After that, the filters were washed with distilled water and subsequently dried.

## 5. Conclusions

This study showed that the PS80 in Blincyto^®^ was not associated with leakage from the filter membrane or loss of mechanical strength. Leakage during long-term administration with portable infusion pumps was hypothesized to be caused by filter blockage and protein aggregation, suggesting that temperature plays an important role.

## Figures and Tables

**Figure 1 ijms-24-05729-f001:**
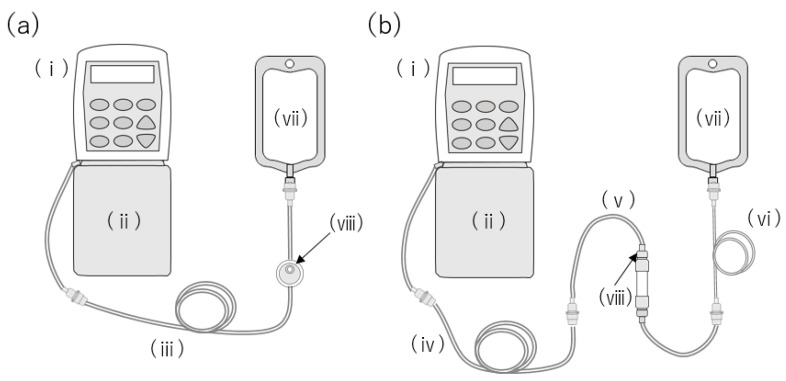
Administration apparatus for intravenous blinatumomab therapy at home. (**a**) CADD Legacy^®^ PCA pump connected with the CADD^®^ extension set with a 0.2-micron air-eliminating filter. (**b**) Administration route used at St. Luke’s International Hospital.: (i) CADD Legacy^®^ PCA, (ii) CADD^®^ 250 mL medication cassette reservoir, (iii) CADD^®^ extension set with male/male Luer, 0.2-micron air-eliminating filter, integral anti-siphon valve, and clamp, (iv) CADD^®^ extension set with male/male Luer, integral anti-siphon valve, and clamp, (v) JMS^®^ Infusion Filter Set, (vi) TOP^®^ Extension Tube, (vii) empty infusion bag, and (viii) air vent.

**Figure 2 ijms-24-05729-f002:**
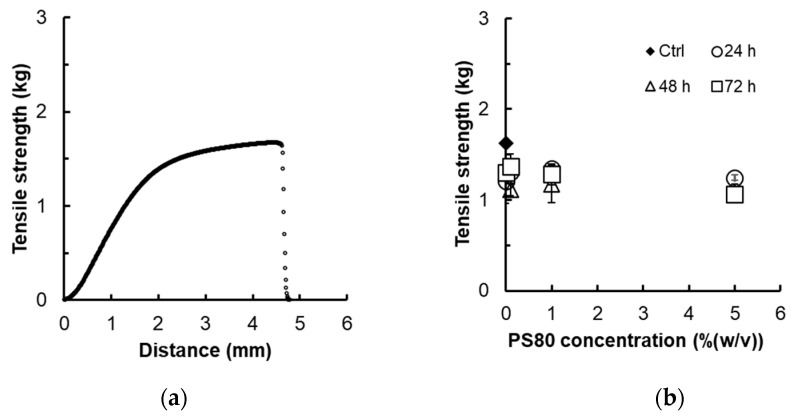
Tensile strength of specimens: (**a**) stress and strain curve of non-treated polyether sulfone (PES) membrane (3.0 × 10.5 cm) and (**b**) breaking strength of polyether sulfone (PES) membrane immersed in the different concentrations of PS80 solution after 24, 48, and 72 h. (*n* = 3).

**Figure 3 ijms-24-05729-f003:**
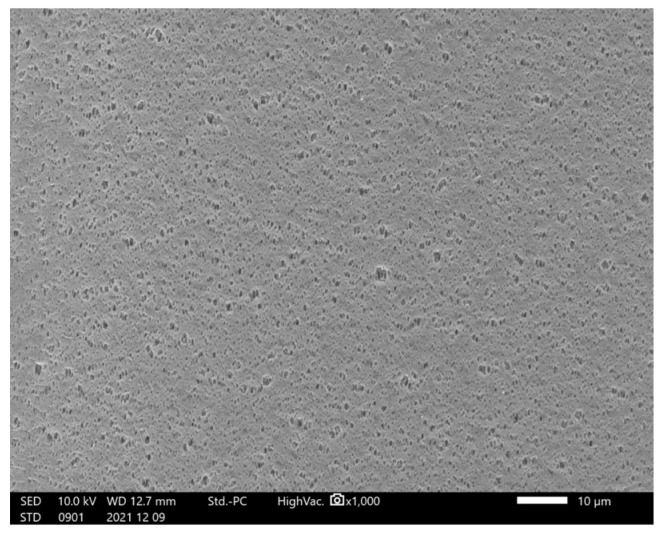
Hollow fiber filter surface after PS80 administration.

**Figure 4 ijms-24-05729-f004:**
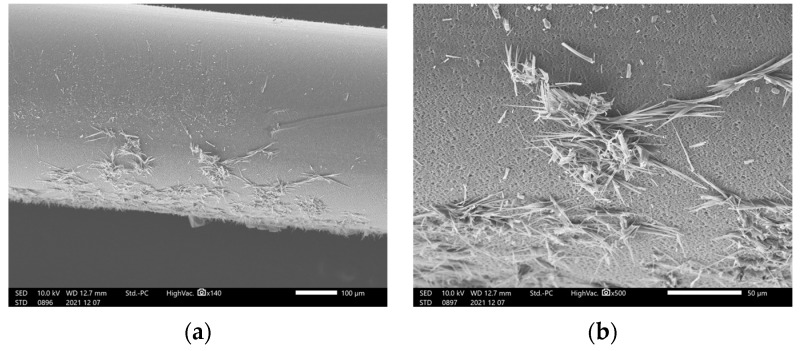
Hollow-fiber filter surface deposits observed during blinatumomab administration: (**a**) low- and (**b**) high-magnification.

**Figure 5 ijms-24-05729-f005:**
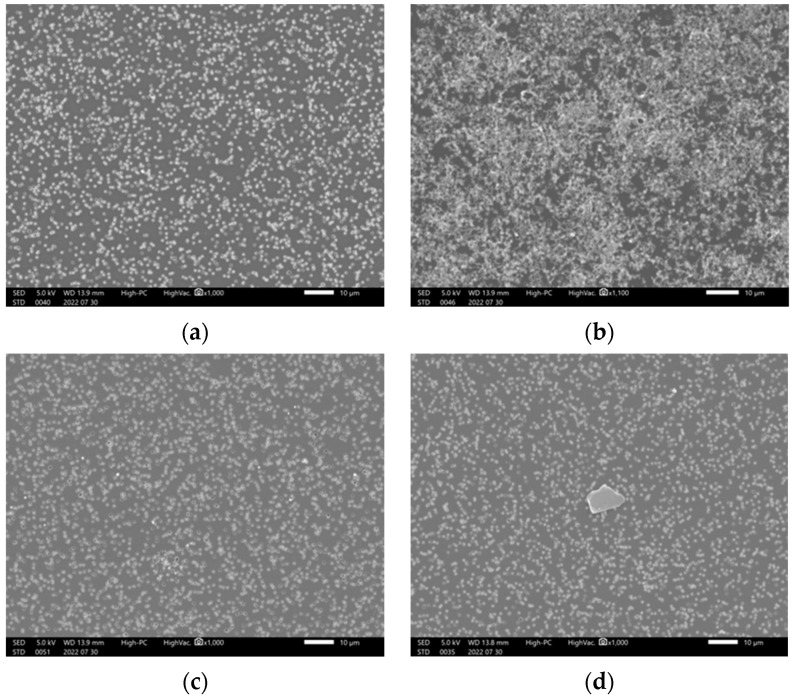
Precipitation observed after subjecting the blinatumomab drug solution to various stimuli (220730): (**a**) treated at 40℃, (**b**) heated to 100 °C, (**c**) after UV light irradiation, and (**d**) after black light irradiation, for 15 min.

## Data Availability

Not applicable.

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
