# Peer review of "SEM Observation of the Filter after Administration of Blinatumomab: A Possibility of Leakage during Home Administration Using a Portable Infusion Pump"

_ijms, 2023, doi:10.3390/ijms24065729_

Round 1
Reviewer 1 Report
Patient comfort during oncological treatment is very beneficial, it has been shown that it improves the effectiveness of therapy. One of the important elements is the possibility of continuing treatment at home, which has been sought in oncology for years.
Home treatment, however, faces many problems and the authors of this manuscript address them. They analyzed the effect of one of the stabilizers used on abnormalities observed during the administration of monoclonal antibodies in the treatment of acute lymphoblastic leukaemia. The results of the observations exclude the adverse effect of the PS80 stabilizer on the course of infusion. The work is interesting, shows a different point of view and factors influencing the effectiveness of therapy than those dependent only on the biology of the disease and the treatment administered.
Author Response
I appreciate for the positive comments.
Reviewer 2 Report
The paper is a bit of a niche area. Difficult for more general readers.
The abstract should be very clear. The authors claim 'we investigated the cause of 14 leakage of blinatumomab', but they only investigate one detail taht could be involved.
I do not understand the sentemce (?) below: 'Scanning electron microscopic observations of anti- 16 body-derived precipitate on the surface of the filters after physical stimulation of the injection solution'
They saw 'no aparrent changes' so why 'physical stimulations should be avoided'.
In general the study would be more interestin if the impact of differnt filters /surfaces /materials were compared. There is just observation but there is no comparsion or control.
This is a major drawback.
Author Response
Comment 1. The paper is a bit of a niche area. Difficult for more general readers.
The abstract should be very clear. The authors claim 'we investigated the cause of leakage of blinatumomab', but they only investigate one detail taht could be involved.
Response: In accordance with the comment, the abstract was revised.
(Page 1, Line 14-15) “Therefore, we investigated device-associated causes of blinatumomab leakage.”
Comment 2. I do not understand the sentemce (?) below: 'Scanning electron microscopic observations of anti- 16 body-derived precipitate on the surface of the filters after physical stimulation of the injection solution'
Response: In accordance with the comment, the sentence was revised.
(Page 1, Line 16-18) “From scanning electron microscopic images, precipitate on the surface of the filters was observed after physical stimulation of the injection solution.”
Comment 3. They saw 'no aparrent changes' so why 'physical stimulations should be avoided'.
Response: We observed no apparent changes to the filter after exposure to the surfactant solutions. But precipitate on the surface of the filters was observed after physical stimulation of the injection solution. Therefore, the sentence was revised as below.
(Page 1, Line 15-16) “We observed no apparent changes to the filter and its materials after exposure to the injection solution and surfactant.”
Comment 4. In general the study would be more interestin if the impact of differnt filters /surfaces /materials were compared. There is just observation but there is no comparsion or control.
This is a major drawback.
Response: As more interesting topics, the effects of filters must be clarified by comparing between many factors of filter. However, we thought that it should be reported as soon as possible. The sentence has been already mentioned as below.
(Page 5, Line 148-151) “Finally, insufficient consideration has been given to the physical stimuli that contribute to precipitation. However, we discovered a filter without shortcomings that is convenient for patients and caregivers in home administration, thus the urgent need to report it in literature.”
Reviewer 3 Report
In this paper, Megumi Takano et al., made a study to investigated the cause of “leakage of blinatumomab, .a bispecific T-cell engaging 10 (BiTE) antibody and is approved for the treatment of relapsed/refractory acute lymphoblastic leukemia
The subject of the article could be relevant relevant as a technical proof of concept related with the material used in the perfusion. But the authors could not translate this result to the administration of blinatumab at home, because there several limitations to be applied in clinical practice at home, contrary to what the authors conclude.
In this context, there are several concerns related with the title, objectives, methodology and conclusions of the work.
First, the title “Factors contributing to leakage of blinatumomab during home-based continuous infusion” are not related with the authors conclusion: “This study showed that the PS80 in blinatumomab was not associated with leakage 193 from the filter membrane or loss of mechanical strength. Leakage during long-term ad-194 ministration with portable infusion pumps was hypothesized to be caused by filter block-195 age and protein aggregation, suggesting that temperature plays an important role.” And this conclusion are the only one the authors can take and they cannot translate it and assume that this is the only problem related with the administration of this kind of drugs at home.
The problems are not only related to the leakage of blinatumumab. And the secondary effects? And I don´t think that secondary effects have a lower probability as the authors mention. There are several secondary effects that coul limit the administration of the drug at home, that can be consult int the approval of the drug by international agencies (FDA and EMA, for instance) and that can be also read (10.1634/theoncologist.2019-0559), namely:
“Treatment‐emergent grade ≥3 adverse events (preferred term in ≥5% of subjects) included neutropenia (15.5%; 18/116), pyrexia (7.8%; 9/116), leukopenia (6%; 7/116), and alanine aminotransferase (ALT) increase and tremor (5.2%; 6/116 for each).Neurologic events were reported in 52.6% (61/116) of subjects. Infections was reported in 41.4% (48/116) of subjects. Cytokine release syndrome (CRS) was reported in 4 of 116 subjects (3.4%). All events occurred in cycle 1. Other adverse events included infusion reaction (82%), drug related hepatic disorders (14.7%), tumor lysis syndrome (6.9%), and decreased immunoglobulins (6.9%). The subject incidence of events suggestive of capillary leak syndrome was 16.4%.”
If blinatumumab is administrared at home, how this problem are solved? and some of them, as CRS, could be fatal.
But, the results obtained by the authors related with capillary leak syndrome may contribute to solved the cause of leakage during long-term administration using a portable infusion pump as they find that “filter blockage and increased internal pressure owing to protein aggregation suggest that temperature plays an important role”. This can solve a technical problem of the administration at hospital, but they cant not conclude that this could allow the administration of blinatumumab at home.
Author Response
In accordance with the comment, the manuscript was revised.
(Page 1, Line 2, and 19-21) “In conclusion, the findings of this study assist in the safe administration of antibodies using portable infusion pumps, taking into consideration the composition of drug excipients and the choice of filter type and structure. “
Blinatumomab is initiated under hospitalization, and after confirming its safety, the patient is transferred to home.
In accordance with the comment, the manuscript was revised.
(Page 1, Line 30-33) “The package insert for Blincyto® in the U.S. recommends that it be initiated in the hospital to monitor for side effects such as infusion reactions, cytokine release syndrome (CRS), neurotoxicity, and tumor lysis syndrome (TLS) [4].”
UV and black light wattages were modified.
(Page 6, Line 186-189) “After dissolving blinatumomab in 3 mL of distilled water, the following stimulation conditions were applied to the drug solution: an increase in the temperature (40°C 15 min, 100°C 15 min), vibration for 1 min (VORTEX GENIE2, M&S Instruments Inc., Osaka, Japan), UV light irradiation (30 W 15 min), and black light exposure (3 W 15 min).”
Round 2
Reviewer 2 Report
The paper can be accepted at editor's discreetion.
No much interest for general audience.
The title and abstract must though be more accurate and stress what is really studied.
The title would promise other topics than finally seen in the paper.
Author Response
The title of the manuscript was change as follow; "SEM observation of the filter after administration of blinatumomab: a possibility of leakage during home administration using a portable infusion pump."
